# From Primary MSC Culture of Adipose Tissue to Immortalized Cell Line Producing Cytokines for Potential Use in Regenerative Medicine Therapy or Immunotherapy

**DOI:** 10.3390/ijms222111439

**Published:** 2021-10-23

**Authors:** Maria Paprocka, Honorata Kraskiewicz, Aleksandra Bielawska-Pohl, Agnieszka Krawczenko, Leszek Masłowski, Agnieszka Czyżewska-Buczyńska, Wojciech Witkiewicz, Danuta Dus, Anna Czarnecka

**Affiliations:** 1Laboratory of Biology of Stem and Neoplastic Cells, Hirszfeld Institute of Immunology and Experimental Therapy, Polish Academy of Sciences, 53-114 Wroclaw, Poland; maria.paprocka@hirszfeld.pl (M.P.); honorata.kraskiewicz@hirszfeld.pl (H.K.); aleksandra.bielawska-pohl@hirszfeld.pl (A.B.-P.); agnieszka.krawczenko@hirszfeld.pl (A.K.); danuta.dus@hirszfeld.pl (D.D.); 2Regional Specialist Hospital, Research and Development Centre, 51-154 Wroclaw, Poland; leszek.maslowski@interia.pl (L.M.); acbuczynska@gmail.com (A.C.-B.); witkiewicz@wssk.wroc.pl (W.W.); 3Faculty of Physiotherapy, University School of Physical Education, 51-612 Wroclaw, Poland

**Keywords:** adipose tissue origin mesenchymal stem/stromal cells, HATMSC1, immortalized cell line, cytokine production

## Abstract

For twenty-five years, attempts have been made to use MSCs in the treatment of various diseases due to their regenerative and immunomodulatory properties. However, the results are not satisfactory. Assuming that MSCs can be replaced in some therapies by the active factors they produce, the immortalized MSCs line was established from human adipose tissue (HATMSC1) to produce conditioned media and test its regenerative potential in vitro in terms of possible clinical application. The production of biologically active factors by primary MSCs was lower compared to the HATMSC1 cell line and several factors were produced only by the cell line. It has been shown that an HATMSC1-conditioned medium increases the proliferation of various cell types, augments the adhesion of cells and improves endothelial cell function. It was found that hypoxia during culture resulted in an augmentation in the pro-angiogenic factors production, such as VEGF, IL-8, Angiogenin and MCP-1. The immunomodulatory factors caused an increase in the production of GM-CSF, IL-5, IL-6, MCP-1, RANTES and IL-8. These data suggest that these factors, produced under different culture conditions, could be used for different medical conditions, such as in regenerative medicine, when an increased concentration of pro-angiogenic factors may be beneficial, or in inflammatory diseases with conditioned media with a high concentration of immunomodulatory factors.

## 1. Introduction

According to the up to date (June 2021) US NIH database [1], around 1200 clinical studies on Mesenchymal Stem/Stromal Cells (MSCs) are registered. In experimental therapy, both autologous and heterologous MSCs are applied directly after separation or after being cultured to obtain a larger number of cells. MSCs obtained from bone marrow, adipose tissue, umbilical cord or other sources can be applied to the patient locally or intravenously; therefore, the results of experimental therapies are difficult to compare [2,3,4].

The high interest in MSC application in therapy is due to the relative ease of cell isolation from tissues and the simple method of their multiplication. Additionally, in some tissues, e.g., adipose tissue, a high concentration of MSCs may be found [5]. As MSCs are characterized by the lack of HLA-DR antigens, both autologous cells and cells from unrelated donors can be used for transplantation and all the tedious and expensive process of selecting a donor is omitted. MSCs do not express any specific marker; therefore, according to the International Society for Cellular Therapy (ISCT) definition for their reliable phenotype identification (CD73^+^, CD90^+^, CD105^+^ and CD45^−^, CD34^−^, CD14^−^ or CD11b^−^, CD79-alpha^−^ or CD19^−^ and HLA-DR^−^), plastic adherence and abilities to differentiate into three cell types should be demonstrated. Recently, criteria for MSCs definition were updated by ISCT to include, among others, their tissue origin and functional assays to define their possible mode of action [6,7].

MSCs are administered to the patients mainly for the following three purposes: to enhance tissue/organ repair in regenerative medicine, e.g., in case of chronic wound healing and/or cardiovascular repair after infarction, to modulate the immune system of patients with inflammatory diseases such as GvHD or rheumatoid arthritis and expecting their differentiation into bone or cartilage cells in case of osteogenesis imperfecta and/or osteoarthritis [8,9,10,11,12]. 

There have also been attempts to use MSCs in cancer therapy. Due to the differential effects of the factors produced by MSCs on different cells, both anti-cancer and pro-cancer effects have been observed [13]. Recently, by the lack of effective treatment of the viral COVID-19 disease, attempts have been made to use MSCs in the experimental anti-viral treatment. In patients with moderate/severe ARDS (acute respiratory distress syndrome), retaining the MSCs in the lung and releasing the active factors were proven to be beneficial. Treatment effects, as assessed by the percentages of survival of the group treated with MSCs and the control group, were 83% vs. 12%, respectively [14].

After an intravenous (IV) injection to patients, the number of MSCs drops dramatically after one hour. An intramuscular (IM) administration seems to be the most effective way to keep MSC cells alive. Adult and neo-natal human MSCs can even survive 5 months at an (IM) implantation site when injected into athymic Balb/c ^nu/nu^ mice. The limitations of MSCs’ effectiveness also result from the fact that MSCs lose their secretory properties with the increase in in vitro passages, with the limitation of cell activity resulting from the necessity to freeze the cells before transport or the IBMIR reaction (an inflammatory reaction resulting from the contact of MSC cells with human serum) [15,16,17].

If cells are not given to patients for differentiation purposes, the mechanism of action of MSCs is believed to depend mainly on the following biologically active factors: cytokines, chemokines and growth factors and nucleic acids produced and secreted by these cells and their paracrine effect. The above-mentioned factors can be released into the environment in free form but also as membrane-covered structures—extracellular vesicles (EVs). Their contents and activity, similar to directly secreted factors, are also considered as mechanisms of MSC interaction with neighboring and distant cells. One of the first reports describing a panel of biologically active factors produced by human, native, myeloid MSCs was a study performed by CW Park, demonstrating the high production of only six cytokines out of 120 tested using the antibody array method. Recently, the proteomic analysis, using liquid chromatography-mass spectrometry (LC-MS/MS) revealed above 700 proteins in secretome [18,19].

MSCs and the factors they produce can exert pleiotropic effects, the phenomenon in which a single factor affects two or more distinct, often seemingly unrelated features. These factors can influence other cells in various, even the opposite way, both by supporting their survival and/or stimulating their proliferation, but also by inducing cytostatic or cytotoxic effects. The former may be beneficial in cell regeneration, the latter in tissue remodeling and cancer treatment. The influence of these factors on the mobility of other cells and their adhesive properties have been also described. Particular attention is paid to the direct and indirect immunomodulatory potential of factors secreted by MSCs, which could be effective both in regenerative medicine and inflammatory diseases. Interestingly, MSCs under the influence of immunomodulatory factors can change both their secretory profile and transform into cells capable of processing and presenting antigens regulating adaptive immunity [20,21,22,23].

Despite the positive results obtained in many clinical trials in Europe, only one therapy with allogeneic MSCs for Perianal Crohn’s Disease (Alofisel) has been approved for use at present [24]. 

Taking into consideration that the therapeutic effect is caused mainly by soluble factors, it seems rational to replace MSCs in regenerative medicine and inflammatory diseases therapies with the factors they produce [25,26,27]. Single agents with potential wound healing effects have been also tested. Therefore, over 20 years of Japanese bFGFu research in wound healing has recently been assessed as still incomplete to be assessed indisputably [28]. In Cuba, an attempt has been made to use EGF in wound healing, paying special attention to in vivo experiments of the parenterally administered EGF on epithelial and nonepithelial organs in terms of mitogenesis and cytoprotection [29].

In line with our hypothesis about the possibility of replacing MSCs with the factors they produce, we established an immortalized MSC line derived from adipose tissue to produce conditioned media rich in growth factors that could be administered to patients [30]. We assume that a stable, immortalized cell line that is not undergoing the aging process should be a more effective producer of biologically active factors than primary culture cells. Additionally, an immortalized cell line can produce practically unlimited amounts of biologically active factors, enabling their use in therapy in a repeatable and optimizable manner regarding the dose and the number of administrations. Unlike the recombinant growth factors used in the clinic, which are usually produced by bacterial or mammalian cells, immortalized MSCs cell lines glycosylate factors in a manner characteristic of humans, which makes them more stable [31]. For this purpose, two cell lines, one derived from human adipose tissue and one from bone marrow, were prepared, and the description of culture methods and the composition of supernatants have been patent pending.

One of the cell lines—Human Adipose Tissue Mesenchymal Stem/Stromal Cells 1 (HATMSC1) cell line—presented in this publication was at first phenotypically characterized in detail. When its phenotype was found to be consistent with the MSCs’ definition, the cytokine production by this lineage was assessed in comparison with primary cell cultures derived from patients using a semi-quantitative antibody matrix method and appeared to be higher compared to the primary cultures of cells.

As MSCs can change their secretory profile under the influence of various factors in the environment, the conditioned medium produced by the HATMSC1 cells line grown under various culture conditions was prepared and assessed in terms of the content of active factors. For the basic conditioned medium obtained from the optimal cell density, under normoxic and serum-free conditions, the regenerative and pro-angiogenic potential in vitro was presented. For conditioned media obtained after HATMSC1 stimulation, we expect even higher activity but tests to assess immunomodulatory activity must be prepared and optimized. 

## 2. Results

### 2.1. Phenotypic Analysis of Freshly Isolated MSCs and 10 Days “In Vitro” Cultured MSCs Obtained from Adipose Tissue of Patients with Non-Healing Wounds

The presence of two antigens—CD34 and CD271—considered to be “stemness”-related and three antigens—CD73, CD90, CD105—used to define MSCs was evaluated using flow cytometry on cells in samples separated from the adipose tissue of patients with non-healing wounds with the use of automated system, CELLUTION 800. The results are presented in Figure 1. In the left column, percentages of marker-positive cells found on day 0 and 10 are presented, while in the right column median fluorescence intensities (MFI)—expressing the level of a marker tested on day 0 and 10—are presented. 

In each sample of cells isolated from adipose tissue, relatively high percentages of CD34^+^ cells were found on day 0 (above 75%) and these percentages remained practically unchanged until day 10, whereas CD34 antigen expression levels were down-modulated, more than 20 times, from MFI 1260 on day 0 to 55.5 on day 10. The average percentages of CD270^+^ cells remained similar during the first days of culture (30%); however, their MFI decreased from 70 to 30. During the first days of culture, the average percentages of CD90^+^ and CD73^+^ cells increased from an initial 82 and 71% to 95 and 90%, respectively. The expression level was diminished by half in CD90^+^ cells and remained practically unchanged in CD73^+^ cells. Different results were obtained for CD105^+^ cells. On day 0 only 22% of the cells expressed this marker, but on the 10th day of culture, it was present in above 90% of the cells, being clearly upregulated from 50 to 250 MFI. These results illustrate the plasticity of the MSCs’ phenotype well and confirm the presence of the CD34 antigen on MSC cells derived from human adipose tissue.

### 2.2. Secretion of Cytokines by Primary MSC Cells Derived from the Adipose Tissue of Patients with Chronic Wounds

The secretion of cytokines was evaluated for primary cultures of a patient’s cells. The evaluation was carried out in supernatants prepared from early (2–3) cell passages for the panel of 50 cytokines. Cytokines found in primary cultures of adipose tissue MSCs are presented in Figure 2 as one heat map, where the green color represents a low level and the red color a high cytokines level. The similarity of the produced cytokines profile is noteworthy, although the level of cytokine production varies. In all the samples, the following six cytokines were found: IL-8, MMP3, MCP1, GRO, TGFβ and TIMP1. In most samples, Angiogenin, TIMP2, Angiopoietin 2, IP-10 and uPAR were also detected.

### 2.3. Phenotypic Analysis of Immortalized HATMSC1 Cell Line

The samples of cells remaining after autotransplantation were used to establish an MSC cell line. Cells obtained from three patients were treated with a pSV3 plasmid carrying complete SV40 early region of the large T-antigen gene and the neo^R^ gene using ViaFect as a transfecting agent. After selection with Geneticin, only one attempt succeeded. These cells were treated for a second time by a pBABE-puro-hTERT plasmid and selected with puromycin. The immortalized HATMSC1 cell line was obtained from Patient No. 25, who was being treated for a venous leg ulcer. The antigens detected on the immortalized HATMSC1 confirming the phenotype of MSCs are presented in Figure 3. According to the ISCT requirements for MSCs, the cells express CD73, CD90 and CD105 antigens. After immortalization, the cells lost expression of the CD271 marker, while the CD34 marker remained on the cell surface, although its expression was low. As with typical MSCs, the cells of the human HATMSC1 cell line express HLA ABC antigens, but no HLA DR antigens, nor a hematopoietic cell marker (CD45) and an endothelial cell marker (CD31). It is also possible to show a low expression of the stromal cells’ marker, Stro-1, which is considered by some researchers to be a marker of MSCs and CD140a on the cell surface. In addition, the cell markers considered to be typical for pericytes can be shown on HATMSC1 cells. These are among others such as CD140b, GD2, NG2, CD13, Nestin and adhesive molecules, MCAM/CD146. The cells also express other markers related to adhesion properties such as ICAM-1/CD54, VCAM-1/CD106, ALCAM/CD166 and HCAM/CD44. Interestingly, the expression of ICAM-1 and VCAM-1 increases about 10 times after stimulating the cells with immunomodulatory factors (TNF and INF), as presented below. Of the 25 markers shown, the HATMSC1 cell line does not express HLA-DR, CD45 and CD31. On the other hand, a low expression of the CD34, Stro-1, VCAM-1 and SSEA1 antigens was assessed as statistically significant on the basis of the Kolmogorov–Smirnov statistics. The D-values for all these markers exceed the required significance threshold of 0.2 with *p* ≤ 0.001 (D = 0.57, D = 0.55, D = 0.53, D = 0.66, respectively).

Finally, the presence of markers related to stemness or self-renewal, e.g., stage specific antigen SSEA-1 and especially SSEA-4, and the presence of the transcription factor OCT3/4, which is critical for maintaining embryonic stem (ES) cells in a pluripotent state, and Sox2, one of the key transcription factors that play an essential role in maintaining pluripotency of stem cells, were demonstrated.

### 2.4. Comparison of the Secretory Profile of Primary Cells and the Immortalized HATMSC1 Line

The production of biologically active agents by the immortalized cell line HATMSC1 and by primary cells was compared. In Figure 4, only the factors produced by cells derived from at least three of the five patients tested are presented. The last bar representing factors secretion by the immortalized cell line was the highest. The concentrations of Angiogenin, Fractalkine, MCP-1, GRO, TGF, TIMP-1 ana and TIMP-2, Angiopoietin and uPAR achieved even nine times higher levels in supernatants produced by immortalized cells. IL-8 secretion was an exception and appeared to be produced to a higher level by primary culture cells than by an established line. It should be noted that the experiments with immortalized cells have been repeated four times (and data scattering was small), whereas samples of cells obtained from the patients were not enough to repeat the estimation. 

### 2.5. Cytokine Production as a Function of the Initial Cell Density 

For a more precise evaluation of the active agents produced by the immortalized HATMSC1 cells, another protein array was used to assess the production of 120 proteins. Assuming that the production of biologically active factors may depend on the initial cell culture density, HATMSC1 cells were cultured in three simultaneously established cultures with an increasing initial density of 1.4 × 10^4^, 2.4 × 10^4^ and 3.3 × 10^4^ cells plated per cm^2^ area of the culture vessel. The optimal starting culture density (in which the highest concentration of the agent is produced) for over 50 factors turned out to be an intermediate density (2.4 × 10^4^ cells/cm^2^). As presented in Figure 5, several factors, e.g., Acrp30, AgRP, Angiopoietin 2, FGF-9 or G-CSF, were produced at the highest concentration under these conditions. Only a few factors, including the cytokines ENA-78, IL-17 or IL-8 were produced at a higher concentration in proportion to the culture density. Surprisingly, another 15 factors were also found, for example, IL-6, TNF-β and Osteoprotegerin, that were produced in higher concentrations by the lower density cultures. Thus, in further studies, an intermediate number of cells was used to set up a culture.

It is noteworthy that the cytokines produced by HATMSC1 have a broad spectrum of target cells (endothelial cells, fibroblasts, keratinocytes, lymphocytes T, B, NK, macrophages) and assuming the detection threshold of 10%, we have only not detected three factors (IL-2, MIG and BTC in the supernatant), and a few are at the limit of detection. Additionally, many factors produced by the HATMSC1 cell line, as detected by a protein array for a panel of 120 proteins, were not detected in a supernatant of primary cell cultures detected using a protein assay for 50 proteins. 

The results for selected cytokines are shown in Figure 5. All the results for the panel of 120 factors are included as Appendix A.

The secretion of cytokines by the HATMSC1 line was evaluated using a RayBio^®^ C Human Cytokine Antibody Array for a panel of 120 cytokines and expressed as a histogram relative to a positive control. Cell cultures were established starting from the following densities: 1.4 × 10^4^, 2.4 × 10^4^ and 3.3 × 10^4^ cells plated per cm^2^ area of the culture vessel. The data represent the mean from a duplicate assessment ±SEM.

### 2.6. Cytokine Production as a Function of Normoxic or Hypoxic Culture Condition

Since hypoxia conditions prevail in the healing wound, it was interesting to see if the cells respond with a change in the level of produced factors during the culture in hypoxic conditions as compared to normoxia. Figure 6a presents the production of selected cytokines by the HATMSC1 line in normoxic and hypoxic conditions. After a 24 h culture in hypoxic conditions, the production of about 20 factors decreased by at least 20%, e.g., EGF, Eotaxin, bFGF, sTNF-R1, TIMP-1 and TIMP-2. At the same time, production of Angiogenin, IL-6 or MCP-1, SCF, SDF-1, GRO, IL-8, MIP-1alpha, MIP-3-beta, uPAR and VEGF increased (1.5–2.5 times). The level of other factors remained unchanged, or the changes were below 10%. It is worth noting that the factors whose level increased under hypoxic conditions are factors with known pro-angiogenic activity. All the results for a panel of 120 factors are included as Appendix A. Additionally, cell surface markers were assessed using flow cytometry after a 24 h incubation in hypoxic conditions; however, no expression changes were noticed (Not shown).

The Antibody Array method is a semi-quantitative method. Therefore, at the beginning of the research, we validated what the actual production level of the selected cytokines is and whether the changes in the level of production assessed by the Antibody Array method correspond to the more precise Multiplex ELISA method. The results are presented in Figure 6b. The tendency for increasing or decreasing the production of selected factors correlates 100% between the two methods. The differences in the cytokine production level in (normoxia vs. hypoxia) assessed using the Multiplex ELISA method are more evident for IL-8, VEGF-A and Angiogenin than by the Antibody Array method. On the other hand, the proportions of increase and decrease in the production of MCP-1 and MCP-3, respectively, are very similar. The production level is expressed in terms of picograms but is higher in the case of IL-8 and VEGF-A and reaches a value of 2–2.5 µg/mL.

### 2.7. Cytokine Production as a Function of Substrate Type and Extended Culture Time

The production of supernatants was carried out in serum-free conditions for 24 h to make sure that cytokines are produced by the tested cells and are not serum derived. As serum free conditions are not optimal for cells, planning an extension of the culture time to 72 h, the culture plastic was additionally changed to collagen discs, which could improve cell viability. As presented in Figure 7, the actual expression level of several cytokines, e.g., PDGF-BB, RANTES or SCF, was higher as cells grew on collagen but at the same time, the expression of other cytokines dropped dramatically, e.g., GM-CSF, PARC, FGF-4 or TPO. In general, as can be seen from the complete presentation of the results, the production of over 60 factors on culture plastic is higher than on collagen, while the production of the other 17 factors is higher when the cells are grown on collagen discs. All the results for a panel of 120 factors are included as Appendix A.

### 2.8. Production of Cytokines in TNF-Alpha Stimulated HATMSC1 Culture

MSCs are known for their ability to respond to microenvironmental factors. The experiment was to determine whether known immunomodulatory factors such as TNF, INF and LPS can also affect immortalized cells and how the production of cytokines changes under their influence.

The profile of secreted active factors changed after a 24 h culture HATMSC1 cells in the presence of 50 µg/mL of TNF-alpha. Out of the 120 examined factors, 80 were produced in a concentration of at least 50% higher than in the control group. All the results for the panel of 120 factors are included as Appendix A, while Figure 8a shows selected immunomodulatory (INF-alpha, IL-15, IL-1 alpha and beta, IL-6, RANTES, TNF-alpha and beta, TGF-alpha and beta, IL-8), chemotactic (Eotaxin, GCP-2, GRO, SDF-1) and growth factors (IL-3, SCF, IL-8, VEGF-D) whose production significantly increased after TNF-alpha stimulation. In contrast, the secretion of IL-7, I-TAC or sTNF-R1 was decreased.

In addition to changing the level of cytokine production, treatment with TNF-alpha caused a change in the phenotype of the MSCs. The expression of adhesive molecules ICAM-1 and VCAM-1 increased significantly, as shown in Figure 8b, which indicates cell activation.

### 2.9. Production of Cytokines in INF Stimulated HATMSC1 Culture

HATMSC1 cells also responded by increasing the production of biologically active factors upon stimulation with INF-gamma in a dose of 50 µg/mL. Out of the 120 factors, the concentration of 70 was at least 50% higher than in the control group. All 120 factors are included as Appendix A, while Figure 9a shows only two cytokines, namely, IL-7 and AcRP30 with a decreased concentration. Similar to the stimulation by TNF, INF stimulation caused an increase in the concentration of immunomodulatory factors (INF-gamma, IL-15, IL-6, RANTES, TNF-alpha and beta, TGF-alpha and beta, IL-8) as well as chemotactic factors (GCP-2, GRO-alpha, SDF-1) and growth factors (IL-3, IL-8, VEGF-D). After INF-gamma stimulation, the concentration of MIG and I-TAC factors in the supernatant increased. This observation strongly suggests that the cell line after immortalization behaves similarly to native cells, as MIG (monokine induced by interferon gamma) and I-TAC (interferon-inducible T-cell alpha chemoattractant) are typical proteins generated by INF-gamma stimulation. After TNF-alpha stimulation, the MIG level rose 3.5 times, but after INF, 22 times, while the I-TAC level was elevated only after INF-gamma stimulation. Interestingly, INF-gamma also influenced the phenotype of cells. In addition to the increase in ICAM-1 and VCAM-1 expression, as in the case of TNF-alpha stimulation, the increase in the expression of the HLA ABC antigen could be observed and the induction of the HLA DR antigen is usually absent on MSCs, as shown in Figure 9b.

### 2.10. Production of Cytokines in LPS Stimulated HATMSC1 Culture

LPS-stimulated cell cultures showed a different cytokine profile than the previous ones. Out of the 120 factors, the concentration of about 40 was at least 50% higher than in the control group, but the production of about 30 factors was reduced. All 120 factors are included as Appendix A, while Figure 10 presents selected factors, with increased and markedly decreased levels. The former are, as in the case of TNF-alpha and INF-gamma stimulation, among others; GCP-2, IL-6, RANTES, TGF-alpha and beta, TNF-alpha and beta, GRO, IL-8, VEGF-D. The production of IL-16, bFGF, FGF-4, FGF-9 and ICAM-1 decreased after LPS stimulation, while the two previous stimulatory factors caused an increase in the production of these factors. IL-7 was the only factor whose production decreased under the influence of all three immunomodulatory factors. In contrast to the treatment with TNF-alpha and INF-gamma, no changes were noticed in the phenotype of the cells treated by LPS (Not shown).

### 2.11. Comparison of the Content of Active Factors Released into the Supernatant of HATMSC1 Line in Free Form and Enclosed in Microvesicles

Active factors produced and secreted by MSCs into the medium exist both in a free and microvesicles form. The higher concentration was found for about 50 factors in the supernatant. The following examples of the factors preferentially released in a free form are shown in Figure 11: Angiogenin, GM-CSF, IL-6, ENA-78, Osteoprotegerin, uPAR and VEGF. About 50 factors out of the 120 tested have a higher concentration in the supernatant. All of them are presented in Appendix A in Appendix A.

However, one can find above 10 factors with a concentration higher in the microvesicles. These are, for example, cytokines BLC, CNTF, Eotaxin, CTACK, GRO, IL-12-p70, the concentration of which is higher in microvesicles: 3× higher in the case of BLC and only slightly (1.15×) higher in the case of IL-12-p70.

### 2.12. Biological Activity of the Supernatant Produced by ATMSC1 Cell Line

A few assays were applied to test and demonstrate the biological activity of the supernatant produced by the HATMSC1 cell line. An in vitro model of wound healing to evaluate the activity was first created. Three cell lines were selected to represent the major cell types involved in wound healing: keratinocytes, fibroblasts and endothelial cells. Using undiluted, 100, 50, 25 or 10% supernatant, it was possible to demonstrate a positive effect of the factors contained in the supernatant, expressed by an increased proliferation of target cells, after 24–48 h in serum free conditions (Figure 12). All three lines proliferated under the influence of supernatant, although fibroblasts and endothelial cells react faster and to a greater extent (2.5- and 3.3-times higher proliferation, respectively, relative to the control) than keratinocytes (a 2.1-times increase). The experiment was conducted for 72 h and at that time the effects of tested supernatants were most evident and statistically significant.

The regulation of the adhesion molecules expression involves the number of cytokines and lymphokines that change the adhesive properties of the cells. Adhesion to the culture plastic of the three main cell types involved in wound healing was observed when keratinocytes (HaCaT), fibroblasts (MSU-1.1) and skin endothelial cells (HSkMEC.2) were plated in serum-free medium, medium supplemented with 10% of the serum or medium supplemented with 50% of the HATMSC1 supernatant. As shown in Figure 13, 1.5–2 h after seeding, the fibroblast and endothelial cells remained fully adhered to the plastic surface and spread on the culture plastic only in the presence of the factors in the tested supernatant.

A tube formation assay can be applied to assess the pro-angiogenic and anti-angiogenic activity of different biological and therapeutical factors. Used in previous experiments, endothelial cell line HSKMEC.2 was plated on the surface of Matrigel, with reduced growth factors, in standard DMEM medium supplemented with 10% of FCS and in the same medium supplemented with 10, 50 and 100% of the HATMSC1 supernatant. As shown in Figure 14a, the best result was obtained when 10% of the supernatant was added to the medium. The network formed by cells growing in the medium with serum was not as complete as in medium with 10% of the supernatant, while cells incubated with 50 or 100% of the supernatant did not form a network at all. The calculated parameters such as mean mesh size, total length and number of nodes were used to precisely describe the effects of cytokines on endothelial cells (Figure 14b). The differences resulting from different cytokine concentrations were assessed with the mean mesh size parameter because this parameter best describes the differences seen in pictures. 

## 3. Discussion

MSCs are a promising tool for regenerative therapies; however, their properties are still not fully understood. These cells do not have one specific marker and are even named differently: MSCs—Mesenchymal Stem Cells or Mesenchymal Stromal Cells or Multipotent Stromal Cells, CFUs-F—Colony Forming Units-Fibroblastic, SVF—Stromal Vascular Fraction, MAPC—Multipotent Adult Progenitor Cells, each of which describes a different aspect of these cells. Recently, A Caplan, author of the term MSCs, suggested changing the meaning of MSCs to Medicinal Signaling Cells, to better describe the properties of these cells [32].

Starting the study with MSCs separated from the adipose tissue of the patients with non-healing wounds for autological transplantation, our attention was drawn by a highly significant negative correlation (*p* < 0.0001) between the wound size and wound closure degree and by the high percentage (on average 75%) of CD34^+^ cells among the isolated cells [33]. Double positive CD34^+^ and CD90^+^ cells were found in different subpopulations of Adipose-derived Stem Cells (ASC), Endothelial Progenitor Cells (EPC) and Vascular Smooth Muscle Cells (VSMC) [34]. The definition of MSCs assumed that they are CD34 negative; although, recently, it has been accepted that this marker is absent in bone marrow-derived MSCs, while it is found in adipose-derived MSCs [6,7,35].

Preparing the cell lines for research, the cells previously used for administration to patients were characterized first. Cell samples from several patients were tested for the percentage of cells expressing the selected markers related to their stemness and MSC phenotype, and their fluorescence intensity. The test was performed on the day of the sample collection and after 10 days of cell culture. The percentage of the following three markers defining MSCs: CD90, CD73 and CD105, was augmented during a short culture, even to 90–99%, probably as a result of MSCs multiplication, and their expression level remained virtually unchanged or even grew, as in the case of the CD105. The most interesting finding is that MSCs from adipose tissue at the time of isolation present high expression of the CD34 marker, which drops dramatically during the first days of culture (MFI fell from 1260 on day 0 to 55.5 on day 10). This explains why many researchers have not found the marker, especially if the cells were evaluated for a CD34 antigen after a long culture. The loss of the CD34 antigen during the culture is not exceptional, hematopoietic stem cells or endothelial stem cells behave similarly [36].

For five of the eight (5/8) patients’ cells proliferating well in the culture, it was possible to evaluate the production of cytokines and growth factors. The most important observation is that in all the samples, six cytokines, namely, IL8, MMP3, MCP1, GRO, TGFβ and TIMP1 may be found. In most samples, Angiogenin, TIMP2, Angiopoietin 2, IP-10 and uPAR were also detected. Therefore, it seems reasonable to state that MSCs from different patients produce quite a similar panel of growth factors. Supernatants are qualitatively similar with some quantitative differences in the level of the produced factors. Similar results were presented by Park et al. where the cytokine secretion profile of native BM-derived MSCs was also shown to be donor-independent [18].

During primary cells culture, their aging occurs; they may also change the number of secreted factors; therefore, a stable cell line with MSCs characteristics seemed a good solution to the problem. According to our experience, double transduction results in the formation of more stable cell lines than a single one. Similar results were presented by L. Balducci et al. [37]. They failed to obtain a line transduced only with the hTERT gene, while the lines obtained by double transduction turned out to be highly proliferating and producing angiogenic factors (HGF and VEGF). Additionally, they characterized their lines in terms of immunophenotype, karyotype and ability to differentiate. They have also demonstrated that lines in in vitro tests are not able to form colonies when growing in agar, which is assumed to be a transformation assay. Similar to the lines shown above, our HATMSC1 line shows numerous changes in karyotype (Not shown). However, the cells of our lineage do not have the ability to create tumors in nude mice, which, in our opinion, proves the lack of neoplastic transformation, but only the immortalization of these cells [30]. None of the approximately 10 lines obtained by us through double transfection have acquired neoplastic properties as evaluated by in vivo tests (Not shown).

After immortalization, the phenotype of the HATMSC1 cell line, obtained from Patient No. 25’s cells, was evaluated to demonstrate the markers necessary to be qualified as an MSC. Similar to the primary cultures, CD90, CD73 and CD105 can be found on cells from the HATMAC1 line. Stemness-related marker CD270 disappeared completely and the expression of CD34 was very low. The cells of the newly created HATMSC1 line expressed, according to the definition, antigen HLA ABC, but did not express HLA DR. Interestingly, the expression of markers considered typical for the pericytes, CD140b, GD2, NG2, CD13, Nestin or CD146, and the presence of adhesion molecules CD146, ICAM-1, VCAM-1, ALCAM and CD44 have been also demonstrated. A handful/few of the markers playing a role in maintaining pluripotency such as SSEA-4 OCT3/4 and SOX2 were also expressed by the HATMSC1 line.

Primary and immortalized cells have been also compared for the production of biologically active agents. As predicted, immortalized cells produce more factors that can be effective in regenerative medicine, especially in wound healing. In particular, large differences are visible in the case of Angiogenin, TIMP1 and TIMP2. In the extreme case of TIMP2, the difference is about ten times in favor of the line. The only exception to the rule turned out to be IL-8 production, which was more effectively produced by the primary cells of all five patients, including the patient whose cells were immortalized. Comparing the results from two RayBio C-series Human Cytokine Antibody Array membranes, detecting a panel of 50 and 120 cytokines, it can be observed that many factors (above 20) not detected in the primary culture of Patient No. 25’s cells, can be found in the supernatants obtained from cell line, e.g., SCF, bFGF, HGF, Leptin, RANTES, G-CSF and VEGF. Most likely, the process of cell immortalization can change the level of cytokines production and stimulate the production of cytokines to which these cells are potentially capable. Therefore, in addition to numerous pro-angiogenic, immunomodulatory and anti-apoptotic factors, growth factors can be found in the supernatant, the main target cells of which are endothelium, hematopoietic cells, liver cells and cells from nervous tissue. This is evident, for instance, in the case of NT-3 (Neutrophin-3, that support the survival and differentiation of neurons). The production of NT-3 by the HATMSC1 cell line reaches up to 70% of the control, while it was not found in the conditioned medium from the primary culture. The use of protein membranes gave us a chance to detect factors that were not expected to be produced by MSC cells, such as Oncostatin or Osteoprotegerin. Out of the 120 tested factors, only three were not detected, but after stimulation with immunomodulatory factors, they did appear in the supernatant. It must be emphasized that these are not all factors produced by MSCs. Therefore, indoleamine 2,3-dioxygenase (IDO) or prostaglandin E2, which are responsible for the immunomodulatory potential of MSCs, were not included in the Antibody Array applied [38].

Due to the underlying cause of the disease, chronic diseases with damage to the skin can be divided into those in which, for various reasons such as age, mechanical damage, damage to skin stem cells, it does not heal nor regenerate, and diseases with a clear autoimmune or inflammatory component, in which the inflammation of the skin cannot be calmed down by properly running homeostatic mechanisms. Therefore, thinking of potential clinical use, we focused on preparing different kind of supernatants. The first one contained an increased concentration of pro-angiogenic cytokines, because one of the fundamental factors for the regeneration of mechanically or radiation-damaged skin is the reconstruction of its vascularization. The second one, should contain an increased concentration of immunomodulatory and chemotactic factors that could bring cells capable of suppressing the inflammatory process, then healing the skin defects and regenerating it to the affected areas.

When examining the factors produced in various culture conditions, it was first observed that the cell density at which culture begins has a significant effect on the production level of various factors. There is no single cell density that is optimal for the production of all the factors. Although most factors are most effectively produced at an intermediate density of the initial culture, a dozen of the factors are produced in higher concentrations when the initial culture is sparse or, on the contrary, dense.

All the subsequent studies were conducted using supernatants prepared from intermediate-density cultures, changing other culture parameters. Literature data and own experience show that under the influence of hypoxia, the secretion of biologically active factors by various cells changes. In hypoxia, EPCs increase the delivery of pro-angiogenic factors while limiting the production of anti-angiogenic factors [39]. Additionally, MSCs can alter the secretion profile under the influence of hypoxia, as demonstrated above. In a publication describing the primary cells, after a 24 h culture under hypoxia, changes in cytokine production are slight [40].

As in the healing wound, the hypoxic conditions prevailed and it seemed to be interesting to check how cells produce biologically active factors in hypoxia, as compared to normoxia [41]. As expected, the cells of the MSCs line increased the production of many pro-angiogenic factors. After 24 h, the changes were not very high (up to 25%), but many factors with proangiogenic activity, such as Angiogenin, IL-6 or MCP-1, SCF SDF-1, GRO, Il-8, MIP-1alpha, MIP-3-beta, uPAR and VEGF were detected in supernatant at a higher concentration; therefore, additive effects may be expected. At the same time, in the supernatant obtained under hypoxic conditions, the level of some immunomodulatory factors was lowered. The studies were also validated with a more precise Multiplex ELISA method. The differences in the level of produced factors assessed using this quantitative method were similar to the results obtained using the Antibody Array method and, in some cases, e.g., for IL-8, VEGF-A and Angiogenin, even more evident.

In another experiment, we were able to show that the cells of the HATMSC1 line after immortalization retained the ability to react to known immunomodulatory factors such as TNF, INF and LPS. Under the influence of all three factors, the production increased by over 50%, respectively, in 80, 70 and 40 factors out of the 120 assessed factors. In the case of hypoxia, the increase in production was usually 10–25%, while under the influence of immunomodulators there were even 11–25-fold increases in the production of certain factors. There is a marked increase in secreted immunomodulatory factors (IFN-gamma, TGF-beta, TNF-alpha and beta, IL-8), chemoattractants (RANTES, SDF1, IL-8) and growth factors (IL-3, VEGF-D, IL-8), although some factors, such as IL-8, are difficult to qualify as have both immunomodulatory, chemotactic and growth factor properties. Interestingly, the production of only a few factors decreases after the stimulation of cells. The only factor that goes down after all three immunomodulatory factors is IL-7—a hematopoietic growth factor, important for B and T cell development. The immunomodulatory effect of INF-gamma and TNF-alpha, not LPS, was also manifested by a change in the phenotype of HATMSC1 cells, consisting of an increase in ICAM-1 and VCAM-1 antigen expression. In the case of INF-gamma, an increased HLA-ABC expression and induction of the HLA-DR antigen could be observed. Interestingly, it is possible to find reports that activation of MSC cells with the mentioned cytokines and LPS may cause various positive therapeutic effects [42,43,44].

Since biologically active factors can be secreted in free or “packed” in a microvesicles form, it seemed interesting to assess whether the factors in both forms are present in equal concentrations or not, especially as, for some time, they are considered potential factors for the so-called Cell-Free Therapy [45]. As expected, the higher concentration of the tested agents was found for about 50 factors in the supernatant. However, one can find above 10 factors, usually with the chemokine function (BLC, CNTF, Eotaxin, CTACK, GRO), with a concentration higher in the microvesicles (MVs), which suggests the active concentration of certain factors in microvesicles. MVs are typically considered as growth factors and RNA carriers for intracellular communication. As carriers of various factors, they can cause various biological effects. In our laboratory, the positive effect of MVs produced by the HATMSC1 line and the HEPC.CB1 endothelial stem cell line on the formation of the endothelial cell network was demonstrated [46]. In contrast, the MVs produced by our second line, HATMSC2, showed a cytotoxic effect on ES-2 ovarian cancer cells and especially OAW-42 [47].

Having obtained the stable HATMSC1 cell line and demonstrated its markers that classify it as an MSC line, the activity of the factors contained in the supernatant was tested. Taking into account the potential of using the produced supernatant in the treatment of, e.g., chronic skin diseases, an in vitro model of skin wounds has been established. The following three cell lines were selected: fibroblasts filling the wound cavity, endothelial cells responsible for blood vessels formation and function and keratinocytes closing the wound and isolating it from the environment. On the other hand, in in vivo experimental therapy using MSCs, skin disease models have the great advantage of being easy to observe and relatively precise to measure and document the effects of the therapy. Therefore, it seems that the model of chronic skin diseases is the most appropriate model for the study of the activity of MSCs and the supernatants they produce [48,49].

In our in vitro model, all of the following three lines: fibroblasts, endothelial cells and keratinocytes proliferated in a serum-free medium only in the presence of supernatants produced by an HATMSC1 line. After 72 h of the test, the greatest increase was observed in the case of endothelial cells and the least proliferated keratinocytes. The 25 and 50% concentration of the supernatant added to the medium was optimal for the proliferation of all the three lines. Interestingly, the 100% supernatant, (without the addition of fresh medium), was sufficient to maintain proliferation activity only for 48 h. It appears that the supernatant does not contain, similarly to serum, all the factors necessary for maintaining constant cell proliferation in vitro and, after a longer period (5–7 days), cell proliferation ceased, even after the administration of an additional portion of the supernatant.

The adhesion of fibroblasts, keratinocytes and endothelial cells, within 0.5–4 h, to culture plastic is perhaps the simplest possible assay to demonstrate the activity of the supernatant factors. After just one hour, cells in the presence of 50% of the supernatant adhered to the plastic and were spread over it. However, this applied only to cells sensitive to the factors contained in the supernatant. As in the case of the proliferation test presented above, endothelial cells and fibroblasts adhered to plastic quickly and flattened on it, while keratinocytes did not. The adhesion of endothelial cells and fibroblasts in the medium with the supernatant was even faster than in the medium with serum.

The last method used to demonstrate the pro-angiogenic activity of different factors was a tube formation assay, assessing the efficiency of network formation by endothelial cells on Matrigel. The mean mesh size, total length and number of nodes were used to precisely describe the effects of cytokines on endothelial cells. In this assay, 100, 50 and 10% supernatants were used compared to the control, which was the serum supplemented culture medium. Network formation by endothelial cells was most efficient in 10% supernatant followed by serum supplemented medium. The other two concentrations of supernatant were too high to stimulate network formation and only caused small clumps to form.

The use of the three methods discussed above was sufficient to reveal the activity of pro-proliferative, pro-adhesive and pro-angiogenic factors in the supernatants produced by the cells of the HATMSC1 line. The next step necessary for a more complex characterization of the supernatants, especially those obtained after stimulation, will be the development or adaptation of in vitro methods showing the potential of the factors as immunomodulatory agents. Undoubtedly, methods of inhibiting proliferation and T lymphocyte function or, conversely, inducing suppressor cells will be considered. The tests with the use of supernatants, affecting monocytes/macrophages and NK cells should not be forgotten either [50,51].

## 4. Material and Methods

All reagents and tissue culture materials used in this study were purchased from Merck, Poznan, Poland Ltd., unless otherwise stated. All reagents for immunostaining were purchased from BD Biosciences, Warsaw, Poland, unless otherwise stated.

### 4.1. Patients

The study involved separated adipose tissue-derived MSCs obtained from 8 patients with nonhealing wounds, treated with MSC autotransplantation in Wrovasc—Integrated Cardiovascular Centre, Wroclaw, Poland, under a research project conducted by Regional Specialist Hospital under the Smart Economy Operational Program. There were 4 males and 4 females, aged from 52 to 82 years (mean age 66.7 ± 10.7). Three persons suffered from diabetes and five from a venous ulcer. MSCs were obtained by liposuction from abdominal fat, enzymatically separated in a closed, fully automated system CELLUTION 800 (Cytori Therapeutics, Austin, TX, USA) and then implanted to the tissue surrounding the wound and to the wound bed. The study protocol was approved by the local Bioethics Committee at the Regional Specialist Hospital, Research and Development Center in Wroclaw, Poland (No. KB/27/2015); all procedures performed in the study were carried out in accordance with the accepted ethical standards compliant with the Declaration of Helsinki. All patients were informed prior, in detail, about the purposes and methods involved in the study, as well as the potential benefits and risks of the therapy, and provided their consent. Small samples of isolated cells were transported to the Institute of Immunology and Experimental Therapy, where the in vitro cultures were established within 3–4 h of cell collection and flow cytometry analysis was performed on the day of cell collection.

### 4.2. Immunostaining

Phenotypes of freshly isolated patients’ cells and cells after 10 days of culture were evaluated using the following PE labeled antibodies: anti CD34, anti CD271, anti CD90, anti CD73 and anti CD105 and an appropriate PE labeled isotype control. HATMSC1 cell line was characterized additionally using PE, FITC or PERCP labelled anti-HLA ABC (Human Leukocyte Antigens ABC), HLA DR (Human Leukocyte Antigens DR), CD45, CD31, STRO-1, CD140a and CD140b, GD2 (Disialoganglioside), NG2 (Neural/Glial Antigen-2), CD13, Nestin, CD146/MCAM (Melanoma Cell Adhesion Molecule), CD54/ICAM-1 (Intercellular Adhesion Molecule-1), CD106/VCAM-1 (Vascular Cell Adhesion Molecule-1), CD166/ALCAM (Activated Leukocyte Cell Adhesion Molecule), CD44/HCAM (Homing Cell Adhesion Molecule), OCT3/4 (Octamer-binding Transcription factor 3/4), SSEA1 (Stage-Specific Embryonic Antigen-1), SSEA4 (Stage-Specific Embryonic Antigen-4) and SOX2 (SRY homology box) antibodies and respective isotypic controls. The last four antibodies, anti OCT3/4, SSEA1, SSEA4 and Sox2, were purchased from Invitrogen, Waltham, MA, USA. For intranuclear or intracytoplasmic proteins, the Perm/Wash reagent was used before staining. After labelling according to the manufacturers’ instructions and careful washing, cells were analyzed by flow cytometry using FacsCalibur, equipped with a 488-nanometer laser. Data were recorded for 10,000 events using CellQuest version 3.3 software. Percentage of positive cells in each sample, expression level of selected antigens and Kolmogorov–Smirnov statistics for each antigen were evaluated. The D values coming from the comparison histogram plots for isotypic control versus specific antibody staining varied between 0 and 1. Values from 0.2 to 1.0 showed significant labelling. Data were presented as histograms using WINMDI 2.8 software.

### 4.3. Cell Culture and Immortalization

Separated adipose tissue MSCs were grown in DMEM medium supplemented with 10% FBS, 1% penicillin/streptomycim solution and L-glutamine in standard culture conditions at 37 °C in 95% air and 5% CO_2_. HATMSC1 cell line was established from cells cultured in OptiMEM medium (GIBCO, Life Technologies, Paisley, Inchinnan, UK) supplemented with 3% FBS in standard culture conditions, transfected with pSV3-neo plasmid, carrying a complete SV40 early region of the large T antigene and marker of G418 resistance www.addgene.org/vector-database/4267/, accessed on 1 December 1999. After selection with geneticine, cells were re-transfected with pBABE-puro-hTERT plasmid www.addgene.org/1771/, accessed on 22 January 2015 and selected with puromycine. Both transfections were conducted using ViaFect^TM^ transfection reagent (Promega, Medison, WI, USA) in serum-free medium, according to the manufacturer’s instructions, as described previously [30].

### 4.4. Conditioned Media Preparation and Growth Factors Production as Evaluated by Antibody Array

Secretory profile of primary patients’ cells vs. cell line HATMSC1 was evaluated in serum free conditioned media. Conditioned media were produced at first by cell line HATMSC1 growing in culture, starting with different cell density (1.4 × 10^4^, 2.4 × 10^4^, 3.8 × 10^4^ cells/cm^2^). Intermediary concentration (2.4 × 10^4^) was selected for further assays. The conditioned media produced by the same cell numbers of primary cells vs. HATMSC1 cells, conditioned media produced by HATMSC1 growing under various aerobic conditions (in normoxia (21% O_2_) vs. hypoxia (1% O_2_)), or cultures on standard plastic vs. collagen sheets were evaluated. Collagen Cell Carrier (CCC) for 24-well plates, were kindly provided by Viscofan Bioengineering, Weinheim, Germany. To mimic pro-inflammatory conditions, cells were stimulated for 24 h by immunomodulatory factors TNFα (50ng/mL), INFγ (50mg/mL) and LPS (*S. typhi*, 50mg/mL). The concentration of growth factors released into the medium and contained in microvesicles (MV) was also estimated after MVs lysis using the lysing agent supplied with the RayBio^®^ test.

Briefly, cells were cultured for 24 h in DMEM medium supplemented with 10% FCS in an experimentally preselected concentration (2.4 × 10^4^ cells/cm^2^). Then, the cultures were thoroughly rinsed with PBS and further cultured in DMEM serum-free medium for 24 h. Collected and centrifuged conditioned media were stored frozen at -20 °C and used for 1 month. Assessment of secreted cytokines was conducted with a semi-quantitative method, with the use of the Custom C-series Human Cytokine Antibody Array for 50 cytokines and then extended with the RayBio^®^ C-Series Human Cytokine Antibody Array C1000 composed of two C6 and C7 membranes, for a panel of 120 cytokines, according to the manufacturer’ instructions. Briefly, the membranes were treated with blocking buffer (30 min, 20 °C). Then, membranes were incubated with 1.5 mL of tested medium overnight (20 h, 4 °C). After careful washing, biotin-conjugated antibodies and a horseradish peroxidase-conjugated streptavidin system was applied to detect immunoreactivity of factors in conditioned medium. Quantification of the reaction was performed using densitometry of X-ray films exposed to chemiluminescence of the membrane. Optical density was determined for each spot as well as for positive and negative controls with the use of Microsoft^R^ Excel-Based Analysis Software Tools. In order to minimize the influence of uncontrolled factors between experiments, each experiment was carried out and compared to a control carried out simultaneously with the test group. Arbitrarily accepted as a positive were proteins/cytokines with production level above 10% of the positive control. Due to the large number of tested proteins, only in selected cases were full names used instead of the commonly used abbreviations.

RayBio^®^ C Human Cytokine Antibody Array is a semi-quantitative method. In order to confirm the levels of active agents produced by the HATMSC1 line, the quantitative method of Human Cytokine/Chemokine Magnetic Bead Panel MILLIPLEX^®^ MAP assay (Merck, Darmstadt, Germany) was used for validation as recommended by the manufacturer with standards and samples, in duplicate. Incubation was performed with shaking at 4 °C (18 h, 750 rpm) and using a handheld magnetic block for the wash steps. Data were obtained on a validated and calibrated MAGPIX^®^ system (Luminex) with xPONENT^®^ software. The median fluorescence intensity (MFI) of the standards, controls and samples were measured and analyzed in Milliplex Analyst software using a five-parameter logistic curve fit method to calculate cytokine concentrations in the samples.

### 4.5. Cell Lines

As target three cell lines, representing main cell types involved in a wound healing process were selected (MSU.1—fibroblasts responsible for the wound cavity filling, HSkMEC.2—endothelial cells for blood vessels formation and function and HaCAT—keratinocytes for closing the wound and isolating it from environment). The cells were cultured in DMEM medium supplemented with 10% FBS, L-glutamine and 1% penicillin-streptomycin and were routinely passaged using a 0.05% trypsin/0.02% EDTA (*w*/*v*) solution (IITD PAN, Wroclaw, Poland).

HSkMEC.2 (human normal skin microvascular endothelial cell line) was established and patented by our research group in cooperation with Prof. C Kieda et al. from Centre National de la Recherche Scientifique, Paris, France, and was used in this study according to the previously described method [52]. MSU-1.1 cell line was obtained through the v-myc oncogene transformation of foreskin fibroblasts [53]. The human keratinocytes cell line (HaCaT) was purchased from the DKFZ collection [54]. HaCaT cells and fibroblast cell line MSU-1.1 was kindly gifted from C. Grillon from Centre National de la Recherche Scientifique, Paris, France.

### 4.6. Capillary-Like Structures Formation Assay

Matrigel matrix with reduced growth factors (BD Biosciences, New Jersey, NJ, USA) was diluted (1:1) in OptiMEM medium at 4 °C, distributed in 96-well microplates in a volume of 50 µL/well and allowed to polymerize at 37 °C for 60 min. Skin endothelial cells HSkMEC.2 were seeded on Matrigel-coated microplates in 100 µL of the DMEM medium supplemented with 10% of FCS (as a control) or in 100, 50 and 10% supernatant produced by HATMSC1 cell line. The direct real-time visualization of the capillary-like structure formation was monitored for 24 h using visible light and the photos were taken 14 h after plating HSkMEC.2 cells on Matrigel. The images were obtained using an inverted Olympus CKX41 microscope equipped with numeric camera linked to a computer driving Stream Start software.

Angiogenesis was quantified by measurement of the mean mesh size, total length of capillary-like structures and number of nodes using ImageJ software. (NIH, Wayne Rasband, USA), as previously described [55].

### 4.7. In Vitro Model of Wound Healing

To evaluate activity of factors produced by HATMSC1 cell line, target cells (MSU1.1, HSkMEC.2 and HaCAT) were cultured in serum free DMEM medium as a control and in tested conditioned media. Cells were seeded in 96-well plates at the density 2 × 10^3^ cells per well and treated with dilutions (10, 25, 50% and undiluted—100%) HATMSC1 supernatants. Metabolic activity was evaluated using the standard MTT test, which is an indirect test to measure cell proliferation, at time 0 and following 24, 48 and 72 h. After a 4 h incubation, 10 µL of MTT stock solution was added to 100 µL of tested wells (conducted in triplicate). After 3 h at 37 °C, supernatant was discarded and 100 µL of solvent (DMSO) was added. Absorbance was read at OD = 570 using Wallac Victor^2^ 1420 multilabel counter (Perkin Elmer, Waltham, MT, USA).

### 4.8. In Vitro Cell Adhesion to a Culture Plastic

Cells involved in wound healing (keratinocytes HaCAT, fibroblasts MSU.1 and skin endothelial cells HSkMEC.2) were plated in various culture media to demonstrate influence of factors produced by immortalized HATMSC1 cell line on the adhesion to a culture plastic. Cells were seeded in 96-well plates at the density of 2 × 10^3^ cells per well in A. DMEM serum-free medium, B. DMEM medium supplemented with 10% of serum, C. DMEM medium supplemented with 50% of HATMSC1 cell line supernatant. Plastic adhesion was monitored for 4 h using visible light and the photos were taken 1.5–2 h after plating.

### 4.9. Statistical Analysis 

Statistical analyses were performed using GraphPad Prism version 7 (GraphPad Software Inc., USA). The data were compared using the one-way ANOVA test with Dunnett’s multiple comparison. All results are presented as mean ± SEM values. Immunostaining data were presented as histograms using WINMDI 2.8 software and evaluated with Kolmogorov–Smirnov statistics for each antigen. The D values coming from the comparison histogram plots for isotypic control versus specific antibody staining vary between 0 and 1. Values from 0.2 to 1.0 show significant labelling. 

## 5. Conclusions

The rapidly proliferating, immortalized HATMSC1 cell line, derived from human adipose tissue, produces biologically active factors at a higher concentration and with greater diversity than the primary cells do. These biological agents could potentially be used in the clinic in place of MSCs. The appropriate method of HATMSC1 line culture allows us to obtain selectively high concentrations of pro-angiogenic factors or high concentrations of immunomodulatory factors.

## 6. Patents

Paprocka M, Kraskiewicz H, Krawczenko A, Bielawska-Pohl A, Maslowski L, Witkiewicz W, Czarnecka A., Czyzewska-Buczynska A, Jedruchniewicz N, Method for the culturing of immortalized human mesenchymal stem cell (MSC) lines, composition of the active ingredients present in the conditioned medium and its application. Polish Patent Office Patent Application No. P.438908, 2021-09-09.

## Figures and Tables

**Figure 1 ijms-22-11439-f001:**
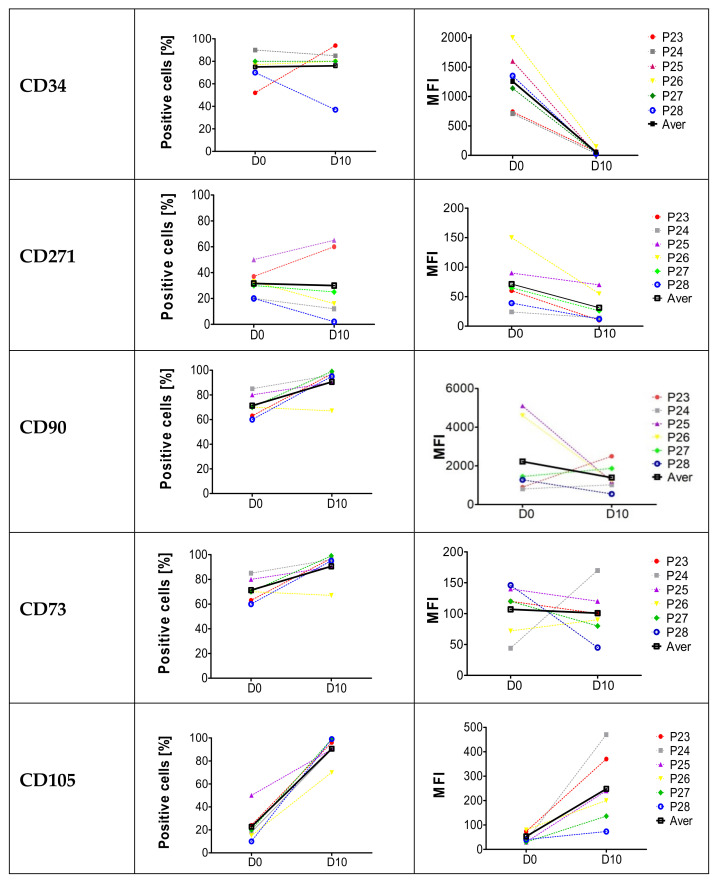
Phenotypic analysis of freshly isolated MSCs and cells after 10 days of culture. Percentages of CD34, CD271, CD90, CD73 and CD105 positive cells and antigen expression level (MFI) in freshly isolated cells and after 10 days cultures were assessed using flow cytometry. The thick, black line represents the average result.

**Figure 2 ijms-22-11439-f002:**
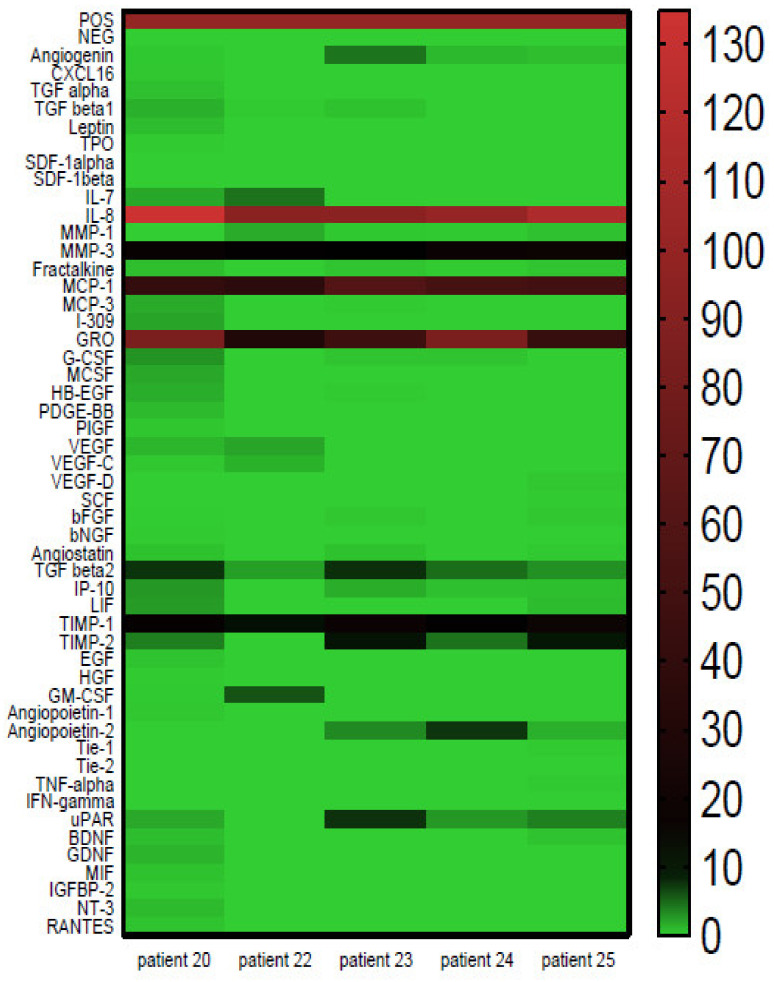
Secretion of cytokines by primary MSCs derived from the adipose tissue of patients with chronic wounds. Secretion of cytokines as evaluated by RayBio Custom C-series human Cytokine Antibody Array for a panel of 50 cytokines. Cell cultures, established with the same cell number, were conducted under serum-free conditions. Cytokine level was expressed as a heat map where green color represents low level and red represents high level of detected cytokines.

**Figure 3 ijms-22-11439-f003:**
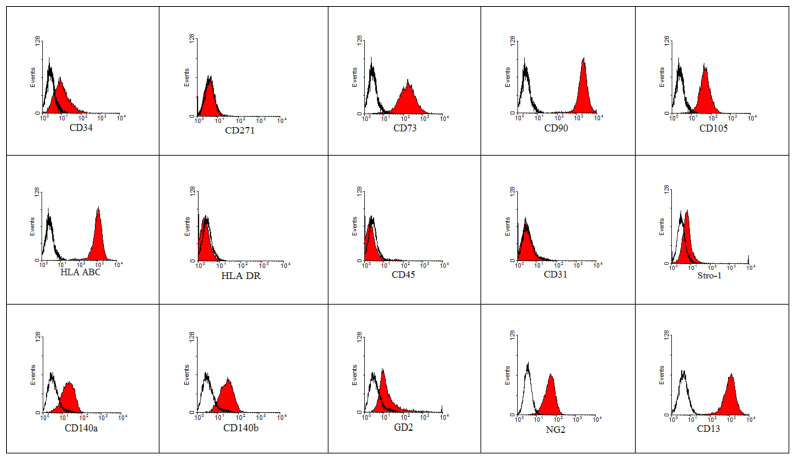
Antigen characteristics of established human HATMSC1 cell line derived from adipose tissue. The mean fluorescent intensity of cells was reported on the x-axis and the number of cells (events) on the y-axis. Black line curves represent isotypic control; red fields represent cells positive for presented antigen.

**Figure 4 ijms-22-11439-f004:**
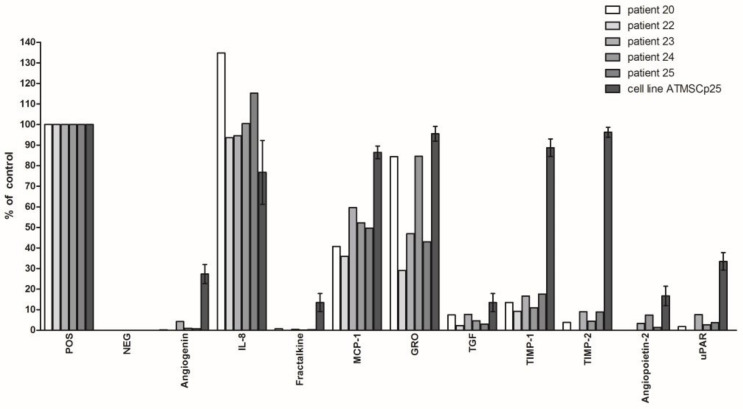
Secretion of cytokines by primary MSCs from the adipose tissue of patients and immortalized HATMSC1 cell line. Secretion of cytokines was evaluated using RayBio^®^ Custom C-series Human Cytokine Antibody Array for a panel of 50 cytokines and expressed as a histogram relative to a positive control. Cell cultures were established with the same number of cells from patients and an established cell line. The data represent the mean from a duplicate of assessment for each patient +SEM from cell line.

**Figure 5 ijms-22-11439-f005:**
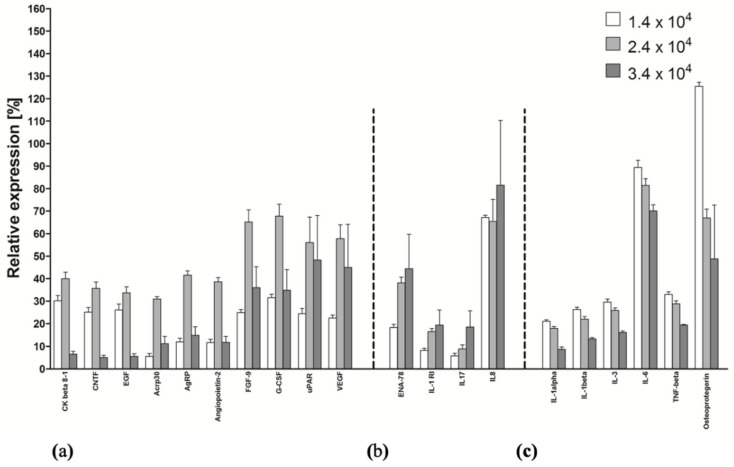
Selected cytokine production as a function of the initial cell culture density. (**a**) Factors produced most effectively by intermediate density cultures. (**b**) Factors produced most effectively by high density cultures. (**c**) Factors produced most effectively by low density cultures.

**Figure 6 ijms-22-11439-f006:**
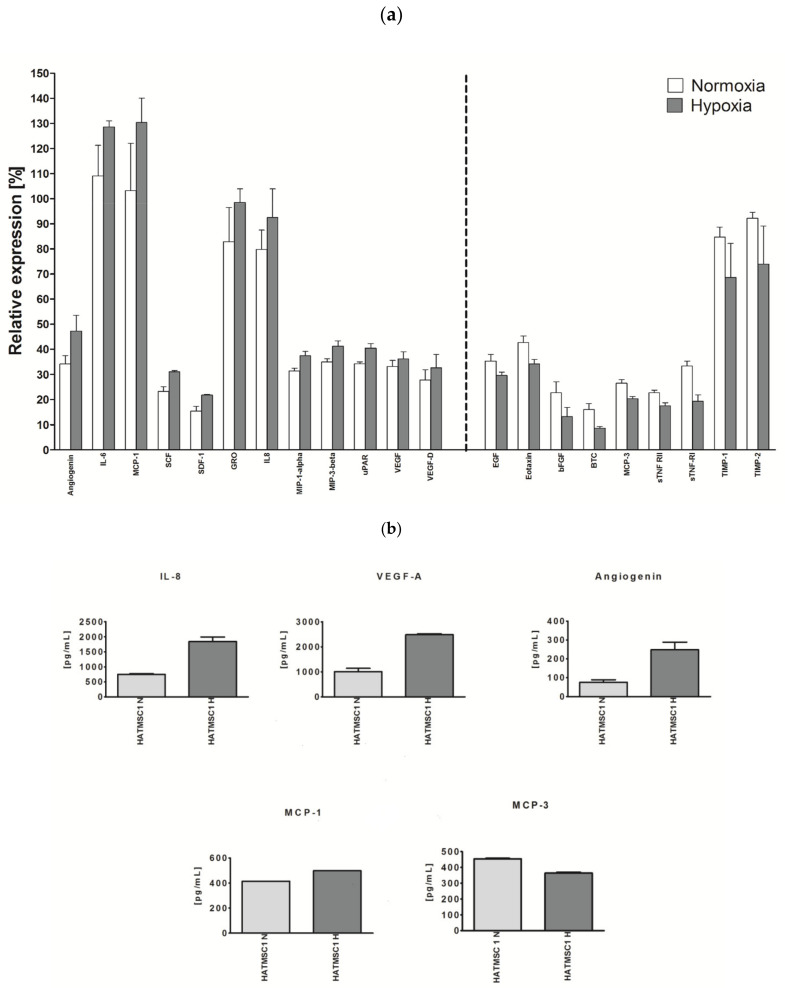
Selected cytokine production as a function of culture under normoxic and hypoxic conditions. (**a**) The secretory profile of the tested line was determined using the RayBio C-series Human Cytokine Antibody Array for a panel of 120 cytokines and expressed as a histogram relative to a positive control. The data represent the mean from a duplicate assessment +SEM. (**b**) Changes in the five selected cytokines levels were confirmed and quantified (for normoxic and hypoxic condition similar to that in **a**) using the Multiplex ELISA method, Human Cytokine/Chemokine Magnetic Bead Panel MILLIPLEX^®^ MAP Kit (Merck, Darmstadt, Germany).

**Figure 7 ijms-22-11439-f007:**
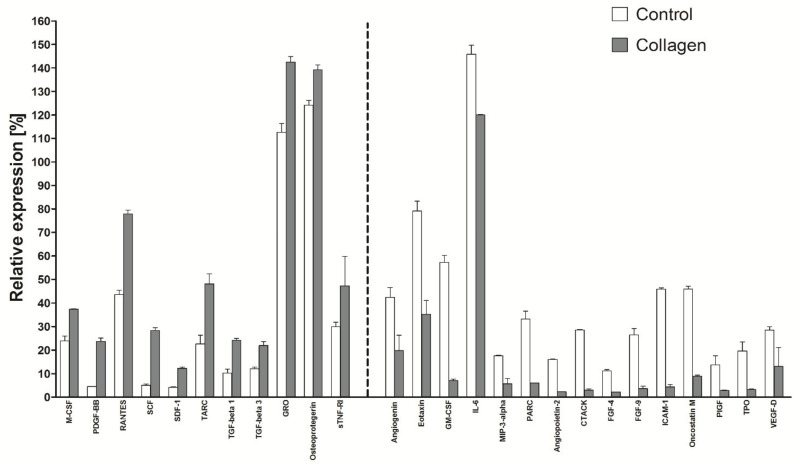
Selected cytokine production as a function of culture on collagen discs and extended culture time. The secretory profile of the tested line was determined using the RayBio C-series Human Cytokine Antibody Array for a panel of 120 cytokines and expressed as a histogram relative to a positive control. The data represent the mean from a duplicate assessment +SEM.

**Figure 8 ijms-22-11439-f008:**
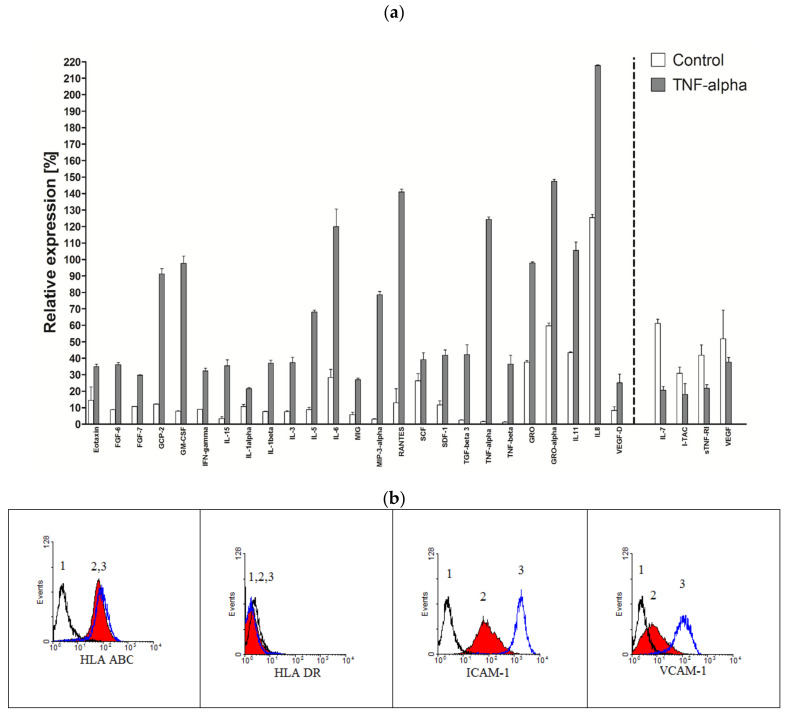
Selected cytokine production after a 24 h TNF-alpha stimulation of cell culture. (**a**) The secretory profile of Table 120, cytokines and expressed as a histogram relative to a positive control. Cell cultures were established and stimulated with 50 µg/mL of TNF-alpha to receive supernatants. The data represent the mean from a duplicate assessment ± SEM; (**b**) Expression of histocompatibility markers and markers related to adhesion on HATMSC1 cells after TNF-alpha stimulation (1—isotypic control, 2—unstimulated cells, 3—cells stimulated with TNF-alpha).

**Figure 9 ijms-22-11439-f009:**
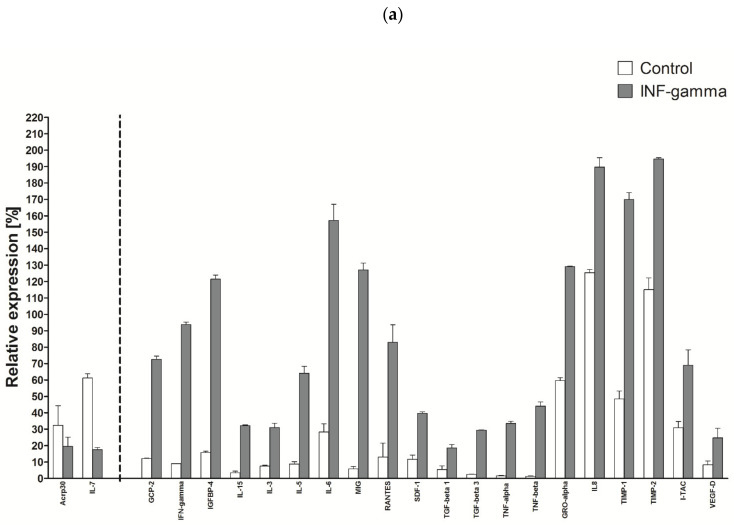
Selected cytokine production after a 24 h INF-gamma stimulation of cell culture. (**a**) The secretory profile of the tested line was determined using the RayBio C-series Human Cytokine Antibody Array for a panel of 120 cytokines and expressed as a histogram relative to a positive control. Cell cultures were established and stimulated with 50 µg/mL of INF-gamma to receive supernatants. The data represent the mean from a duplicate assessment ± SEM; (**b**) Expression of histocompatibility markers and markers related to adhesion on HATMSC1 cells after INF-gamma stimulation (1. isotypic control, 2. unstimulated cells, 3. cells stimulated with INF gamma).

**Figure 10 ijms-22-11439-f010:**
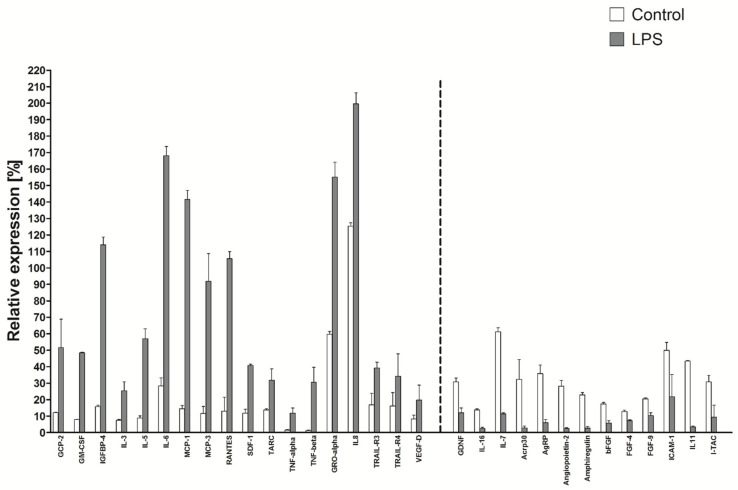
Selected cytokine production after a 24 h LPS stimulation of cell culture. The secretory profile of the tested line was determined using the RayBio C-series Human Cytokine Antibody Array for a panel of 120 cytokines and expressed as a histogram relative to a positive control. Cell cultures were established and stimulated with 100 µg/mL of LPS to receive supernatants. The data represent the mean from a duplicate assessment ± SEM.

**Figure 11 ijms-22-11439-f011:**
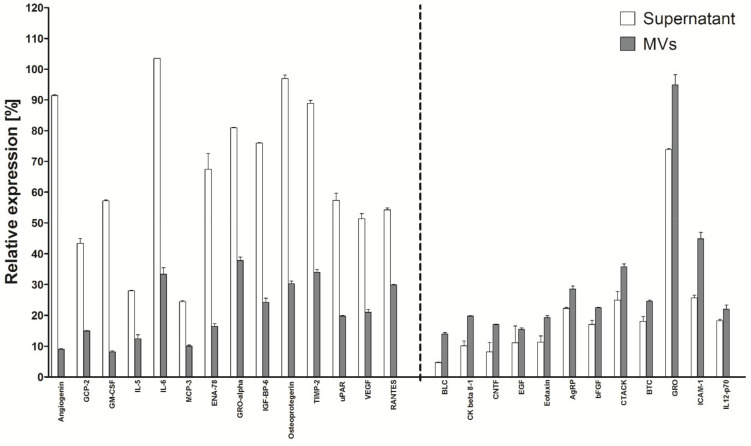
Selected cytokine production in the free form (Supernatant) and in the microvesicles (MVs) form. The secretory profile of the tested line was determined using the RayBio C-series Human Cytokine Antibody Array for a panel of 120 cytokines and expressed as a histogram relative to a positive control. The MVs present in the supernatant were obtained by sequential centrifugations. They were lysed 100% to evaluate cytokine concentration. The data represent the mean from a duplicate assessment ±SEM.

**Figure 12 ijms-22-11439-f012:**
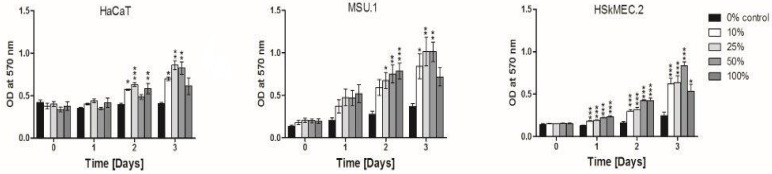
The influence of the HATMSC1 supernatant on the proliferation of cells involved in wound healing (keratinocytes HaCaT, fibroblasts MSU-1.1 and skin endothelial cells HSkMEC.2). Target cell proliferation was evaluated using an MTT test and expressed by OD value at 570 nm after culture for 3 days in the serum free DMEM medium (0% control) and in the presence of diluted (10–50%) and non-diluted (100%) HATMSC1 cell line supernatant. Data represents mean ± SEM, * *p* < 0.05, ** *p* < 0.01, *** *p* < 0.001 from 3 independent experiments.

**Figure 13 ijms-22-11439-f013:**
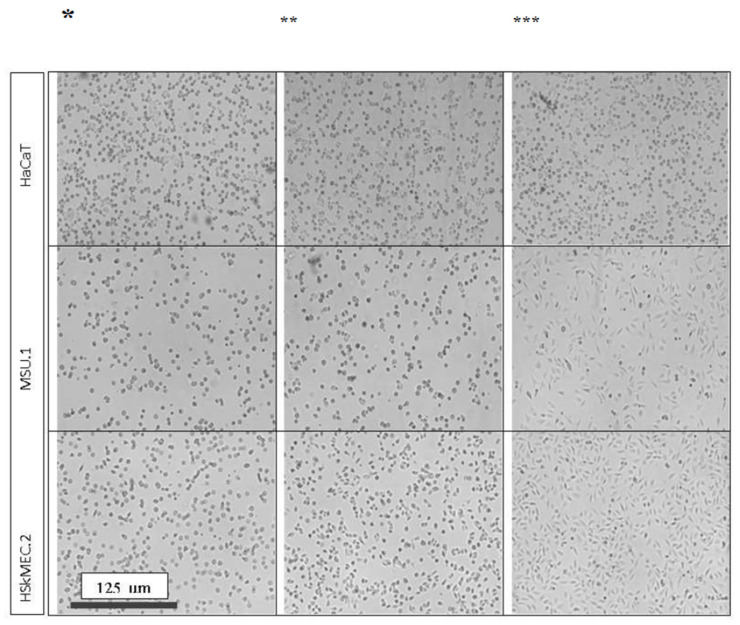
The influence of the HATMSC1 supernatant on the adhesion of cells involved in wound healing (keratinocytes HaCAT, fibroblasts MSU-1.1 and skin endothelial cells HSkMEC.2 to culture plastic). Pictures taken 1.5-2 h after plating target HaCaT, MSU-1.1 and HSkMEC.2 cells show differences in their adhesion to culture plastic in various culture media. * DMEM serum-free, ** Standard DMEM medium supplemented with 10% of serum, *** DMEM medium supplemented with 50% of HATMSC1 cell line supernatant.

**Figure 14 ijms-22-11439-f014:**
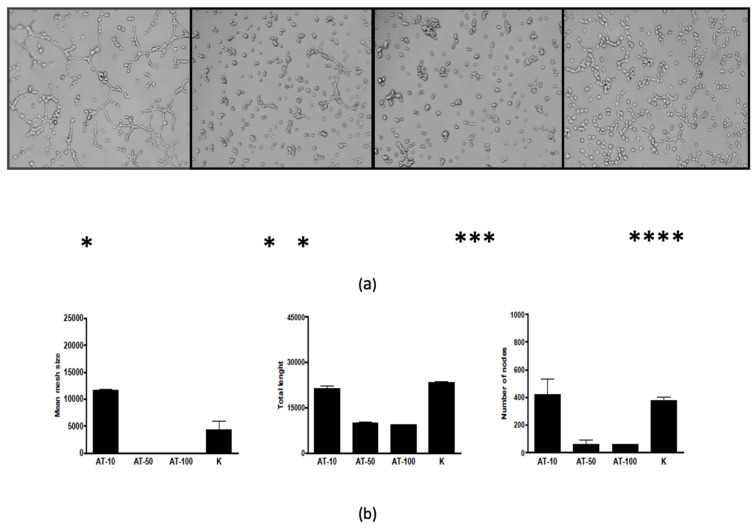
The influence of the HATMSC1 line supernatant on angiogenic activity of skin endothelial cell line HSKMEC.2. (**a**) Pictures taken 14 h after plating HSkMEC.2 cells on Matrigel in DMEM culture medium supplemented with * 10% (AT-10), ** 50% (AT-50) *** 100% (AT-100) HATMSC1 supernatant and **** in with 10% FCS (K). (**b**) Influence of the supernatant on endothelial cells, was evaluated using three selected parameters: mean mesh size, total length and number of nodes. Data represents mean ± SEM of one representative experiment, *n* = 3. *** *p* < 0.001.

## Data Availability

All data generated or analyzed during this study are included in this published article.

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
