# Peer review of "From Primary MSC Culture of Adipose Tissue to Immortalized Cell Line Producing Cytokines for Potential Use in Regenerative Medicine Therapy or Immunotherapy"

_ijms, 2021, doi:10.3390/ijms222111439_

Round 1
Reviewer 1 Report
I think this research is very interesting. There are a lot of information regarding to the supernatant of MSC cell lines.
Potentially, I think this manuscript should be published.
I have a few questions to the authors and a few minor comments below.
The authors described their cell lines as MSC. How about the differentiation ability of those cells?
The authors used the adipose-derived cells from the patients. What if the cells from the healthy people? What kind of benefits are there to use patients cells instead of using healthy people other than availability?
Line 154, it is better to describe MFI in the text. (I found the description in the Figure legend, though.)
Figure 6b No explanation for the columns. May be left columns are normoxia and right ones for hypoxia.
There are several unnecessary spaces between the words.
Reviewer 2 Report
Dear Authors,
Please find the following comments:
1- The title of the manuscript is too long. Please re-write the title and try to shortage it.
2- Please re-submit (Figure 5) and (Figure 14 b) as the resolution of the current figure is not clear.
3- Please check some lines that were highlighted. For example lines 259, 264, 265, 468, and 871
